# Male Pheromones Induce Ovulation in Female Honeycomb Groupers (*Epinephelus merra*): A Comprehensive Study of Spawning Aggregation Behavior and Ovarian Development

**DOI:** 10.3390/cells11030484

**Published:** 2022-01-30

**Authors:** Takafumi Amagai, Daisuke Izumida, Ryosuke Murata, Kiyoshi Soyano

**Affiliations:** 1Institute for East China Sea Research, Organization for Marine Sciences and Technology, Nagasaki University, 1551-7 Taira-machi, Nagasaki 851-2213, Japan; tamagai@nagasaki-u.ac.jp (T.A.); murata-r@nagasaki-u.ac.jp (R.M.); 2Fisheries Technology Institute, Japan Fisheries Research and Education Agency, 116 Katsurakoi, Kushiro 085-0802, Japan; izumida@affrc.go.jp; 3Graduate School of Fisheries and Environmental Sciences, Nagasaki University, 1-14 Bunkyo-machi, Nagasaki 852-8521, Japan

**Keywords:** honeycomb grouper, ovulation, pheromones, spawning aggregation, spawning characteristics, final oocyte maturation, gonadotropin, lunar-synchronized spawning, male–female interaction

## Abstract

This study characterizes the spawning phenomena of the honeycomb grouper (*Epinephelus merra*), which is a lunar-synchronized spawner that spawns a few days after full moon. To elucidate the aggregation characteristics of wild honeycomb groupers, the numbers of males and females at the spawning grounds were counted before and after the full moon. Approximately 20 males were consistently observed at the spawning grounds throughout the study period. Females appeared several days after full moon and rapidly increased in number, peaking four days after full moon (41 individuals). The maturation status of the females aggregating at the spawning grounds was investigated. The gonadosomatic index increased rapidly three days after full moon, and ovulation was confirmed. Individuals with ovulatory eggs were present for three days, after which the number of females at the spawning grounds decreased. Additionally, the role of males in final oocyte maturation (FOM) and ovulation in females during the spawning phase was investigated in captivity. FOM was induced in females reared in water with mature males, suggesting that male pheromones in the water induced FOM via activation of the hypothalamic–pituitary–gonadal axis. This suggests that spawning at the natural spawning grounds was the result of male–female interactions via pheromones.

## 1. Introduction

Most grouper species, some of which are economically and ecologically important, are distributed in the tropical and sub-tropical zones. Groupers are bottom-associated fish that reside in areas with rocky coral bottoms [1]. Coral reef fish, including groupers, exhibit spawning aggregations that can be classified into two types: transient and resident [2]. Large- and medium-sized groupers undergo transient aggregation. They migrate long distances (>2 km) from their normal habitats to their spawning grounds [2]. Conversely, small-sized groupers generally undertake resident aggregation, forming spawning aggregations in their normal habitats, although this phenomenon is not well understood. However, not all small groupers engage in resident aggregation. The white-streaked grouper (*Epihephelus ongus*), which is a small-sized species (total length (TL) < 40 cm), exhibits transient aggregation [3]. The spawning migration of some aggregating species, including some groupers, is synchronized with the lunar cycle, although the timing of aggregation varies even among grouper species, including new moon, full moon, and last quarter moon synchronization [4,5,6].

In groupers, final oocyte maturation (FOM), a process essential to completing meiosis, is thought to occur after migration to the spawning grounds. After FOM, ovulation and hydration occur, and the preparation for spawning is complete. Ovulation patterns, such as duration and frequency, depend on the reproductive cycle and spawning characteristics of a species; in teleosts, the reproductive axis is regulated by external environmental cues, such as water temperature and photoperiod [7,8]. According to Peter and Yu [9], pheromones can also stimulate the physiological changes required for reproduction, including ovulation and spawning in females. Although pheromones may also be important in FOM and spawning induction, their role in maturation and spawning in marine teleosts, including groupers, has yet to be confirmed. In addition, morphological and histological changes during mating have not been reported in wild groupers. This information is important for successful seedling production and the stock preservation of grouper species.

Most reproductive events are regulated by the hypothalamus–pituitary–gonadal (HPG) axis, and in teleosts, two types of gonadotropins (GtH) synthesized in the pituitary, follicle stimulating hormone (FSH) and luteinizing hormone (LH), control gametogenesis [9]. Both FSH and LH are composed of a common α-subunit and a hormone-specific β-subunit [10,11]. GtHs secreted by the pituitary are transported to the ovaries through blood vessels, and gametogenesis is induced via steroidogenesis in the ovaries. In most teleosts, 17α,20β-dihydroxy-4-pregnen-3-one (DHP) regulates FOM. As the steroid hormone that directly induces FOM, it is called the maturation-inducing hormone (MIH) [12]. In some teleost fish, GtHs, especially LH, play a role in FOM via the induction of MIH synthesis and MIH sensitivity in the ovary. The ability to respond to MIH is defined as oocyte maturational competence (OMC) [12,13,14,15,16]. These endocrinological changes during FOM are essential for the successful completion of oogenesis, although there is little information on the endocrine mechanisms of FOM in grouper species. Understanding these endocrinological changes during FOM is key to understanding the baseline reproductive biology of groupers. To elucidate the ovulation and spawning characteristics of groupers, including their endocrine regulation, we used the honeycomb grouper, *Epinephelus merra*, which is one of the smallest groupers (TL < 25 cm). Its reproductive cycle is synchronized with the lunar rhythm during its spawning season (May to August), and oocyte growth peaks are observed at full moon [17]. Our previous work on captive honeycomb groupers revealed that the species spawns continuously for three days, a few days after the full moon [18]. However, details about the spawning of these groupers in the wild are unknown. Although it is thought that honeycomb groupers migrate to their spawning grounds from their resident lagoon at full moon, their spawning characteristics, including their aggregation sites, remain unknown [19]. To address this lack of information, we investigated the aggregation and ovarian development of honeycomb groupers in their natural spawning grounds. Then, we investigated changes in their gonadal development over time, from the initiation of FOM to ovulation. We also investigated the effects of male–female pheromonal interactions on the synthesis of GtHs during final maturation and ovulation in captivity. Moreover, to understand the effect of male–female pheromonal interactions on the induction of FOM, we investigated the response of oocytes to gonadotropin using a human chorionic gonadotropin (HCG) injection.

## 2. Materials and Methods

### 2.1. Animal Welfare

All experimental procedures involving animals were conducted in compliance with the Animal Care and Use Committee of the Institute for East China Sea Research, Nagasaki University, Japan (Permit Number #15-06). All the experimental fish were captured in cooperation with the fishery association of the Nakijin local region. Sampling was conducted as part of a survey to obtain basic information for resource management and this research was scrutinized and approved by the research grant system of the Ministry of Education, Culture, Sports, Science, and Technology of Japan (Grant-in-Aid for Scientific Research #19H03034). According to these guidelines and policies, all experiments were conducted without causing severe distress to the fish. To obtain tissue samples for histological observation, the fish were anesthetized for pain reduction. No other surgical operations or drug administration were performed.

### 2.2. Field Survey of Migration and Aggregation

Observations were performed by snorkeling in the coral reef lagoon at the edge of the Nakijin River, in the northern region of Okinawa Island (26°4159.05 N, 127°5826.17 E) from 27 May to 4 June 2018 (Figure 1). The observation area encompassed the aggregation area (150 m), where aggregation had been observed previously, and included part of the reef edge and the edge wall. The water depth of the observation area was 1–6 m, and the depth of the coral lagoon habitat adjacent to the observation area was 1–2 m. Observations were conducted from 15:00 to 18:00 p.m., and the number of fish in the observation area was counted for 60–90 min per day. The fish were easily identified owing to their distinctive body shapes and fin-tip coloration (Figure 2A,B,D): the females had a swollen abdomen and a yellow fin tip, while the males had a white fin tip. Fish numbers and positions were recorded on a chart to avoid duplicate counting. Additionally, the interactions of males with other males and females were observed from 8:00 a.m. to 19:00 p.m. on 25–28 July 2018 and on 12–23 June and 13–15 July 2019. Reproductive behaviors were recorded as videos or pictures using a STYLUS TG-3 Tough camera (Olympus, Tokyo, Japan).

### 2.3. Survey of Ovarian Development at the Spawning Grounds

To protect the grouper population and allow for future surveys in this field, the number of samplings was kept low. Several females (*n* = 3 or 4 per day) along the reef edge were captured by fishing every day between 30 May and 4 June 2018. It was difficult to use any more individuals in obtaining experimental fish while protecting natural resources. The captured females were transported to Sesoko Station, Tropical Biosphere Research Center, University of the Ryukyus, where they were anesthetized, and their body weight (BW) and TL were measured. The ovaries were removed, and the gonadal weight (GW) was measured. The ovaries were then dissected, and the ovulatory eggs were removed from the ovarian tissue using a spoon. Next, the weights of the ovulatory eggs (OW) and the ovarian tissue containing pre-ovulatory oocytes (PW) were measured separately. The gonadosomatic index (GSI) in non-ovulatory fish was calculated based on the weight of the ovaries according to the following formula:GSI = GW (g)/BW (g) × 100,(1)

In the ovulated fish, the proportions of OW (POW) and PW (PPW) were calculated according to the following formulas, and GSI was calculated as the total of the respective values.
POW = OW (g)/BW (g) × 100,(2)
PPW = PW (g)/BW (g) × 100.(3)

### 2.4. Ovulation and Spawning in Captivity

Experimental fish were captured from 18 to 20 June 2016 (full moon was on 20 June) around Sesoko Island, Okinawa, Japan. The captured fish were transported to Sesoko Station, University of the Ryukyus. The transported fish were divided into males and females by stripping their abdomen to confirm whether they were spermiating or not. Males and females were stocked separately in outdoor 500 L tanks with a shelter for hiding containing natural seawater for several days until the start of the experiment. Two days after the full moon (at 10:00 a.m. on 22 June), the fish were divided into three experimental groups. Females alone were used as controls. Females reared with males constituted the mixed rearing (F + M) group, while females reared alone but exposed to male rearing water constituted the male water (MW) group. Natural seawater was supplied to both the control and F + M group tanks. Male rearing water was continuously supplied from a 250 L tank containing five males to the MW group tanks. The water temperature of all tanks was similar to the ambient marine water temperature (25.9–26.7 °C). The overflow water from each tank was passed through a mesh net for egg collection. The presence or absence of released ovulatory cells was confirmed at 10:00 a.m. every day during the experiment. At this time, buoyant and sunken eggs were collected from the tank using a scooping net to correctly determine egg release. Six females from the control group were sampled prior to the start of the experiment. Then, six females from control and MW groups were sampled at 72, 120, and 144 h after the start of the experiment. In the F + M group, five females were sampled at 72, 120, and 144 h after the start of the experiment. The sampled females were anesthetized using β-phenoxyethanol, and their ovaries were dissected for histological observation.

The MW and control treatments were replicated in the following month. Experimental fish were collected around Sesoko Island during the full moon phase (18–20 July 2016). From 22 to 25 July, the presence or absence of eggs in each tank was confirmed every 24 h (10:00 a.m.) in the control group (7 females) and MW group (7 females), using the method described above. The experiment was censored 72 h after the start of the experiment, when released eggs were confirmed in the MW tank. During this replication, the water temperature was between 26.1 and 27.0 °C.

Until the beginning of the experiment, the fish were fed with the silver-stripe round herring *Spratelloides gracilis* once a day. However, no feeding was carried out during the experimental period.

### 2.5. Histological Observation of Ovaries

The ovaries were cut into pieces, and the tissues were fixed in Bouin’s solution for 48 h. The fixed tissues were dehydrated in a series of ethanol and butanol, and then embedded in paraffin. The embedded tissues were sliced to a thickness of 5 μm and stained with hematoxylin and eosin Y. Based on observations under a light microscope (BX50F4; Olympus, Tokyo, Japan), the developmental stages of the ovaries were classified according to the method reported by Shein et al. [20]. In this study, various oocyte stages, from full vitellogenic to ovulated oocytes, were observed (Figure 3). The FOM oocyte development stages were determined using the following characteristics: During the tertiary yolk stage-I (TY-I), the oocyte was filled with yolk globules and the germinal vesicle was located in the center of the cytoplasm. In the tertiary yolk stage-II (TY-II), the oocyte was filled with yolk globules and oil droplets gathered around the germinal vesicle. In the migratory nucleus stage (MN), the yolk globules and oil droplets coalesced and germinal vesicles migrated to the animal pole. In the ripe stage (R), the coalesced yolk globules were hydrated and the germinal vesicle broke down, and in the ovulatory stage (OV), the egg was released from the follicle. Post-ovulatory follicles and pre-ovulatory oocytes remained in the ovigerous lamella, while the atretic oocytes, oocyte follicle, and zona pellucida were degenerated by apoptosis.

### 2.6. Effect of Pheromonal Cues and HCG on FOM and GtHs Synthesis

Experimental fish were captured during the full moon phase from 7 to 9 June 2017 around Sesoko Island, Okinawa, Japan. The captured fish were transported to Sesoko Station, University of the Ryukyus. Males and females were stocked separately in outdoor 1000 L tanks with a shelter for hiding containing natural seawater for several days until the start of the experiment. Until the beginning of the experiment, the fish were fed with the silver-stripe round herring *Spratelloides gracilis* once a day. However, no feeding was carried out during the experimental period. Two days after the full moon (at 10:00 a.m. on 9 June), the fish were divided into three experimental groups. Females raised alone were used as controls reared in a tank (1 t). Females reared alone and injected with HCG constituted the HCG group. The method of preparation and injection of HCG are described as follows: HCG (ASKA Pharmaceutical Co., Ltd., Tokyo, Japan) was dissolved in saline at the concentration of 2000 IU/mL, then the HCG-dissolved saline was vigorously mixed with cocoa butter (saline:cocoa butter = 1:9, final concentration = 200 IU/mL) for emulsification using a vortex. The HCG solution was kept in 40 °C to prevent solidification until injection. The HCG solution was injected to all females in the HCG group at the start of the experiment. The control group was injected with cocoa butter only. Females reared alone and exposed to male rearing water constituted the MW group. Natural seawater was supplied to both the control and HCG tanks. Male rearing water was continuously supplied from three 250 L tanks containing 6 males to the MW group tanks. The water temperature of all tanks was similar to the ambient marine water temperature (25.1–26.4 °C). At the start of the experiment, females in the control group were sacrificed as initial sampling (6 females), and then every 24 h, sampling in each experimental group (5 females) was continued until 72 h. Sampling was conducted following the method mentioned above. Additionally, the pituitary was sampled and fixed by RNAlater (QIAGEN N.V., Venlo, The Nederland).

### 2.7. Quantitative Real-Time PCR for Analysis of fshb and lhb

Total RNA was extracted from the pituitary using the RNeasy Mini Kit according to the manufacturer’s protocol (QIAGEN, Venlo, Nederland) and quantified using a NanoDrop2000 system. Total RNA (100 ng) was reverse transcribed using the Smarter cDNA synthesis kit (QIAGEN, Venlo, The Netherlands) after priming with random hexamer primers. The transcript abundances of the *fshb* and *lhb* genes were determined by an absolute quantification system using a TaqMan probe quantitative real-time PCR assay. Complete cDNA sequences constructed for honeycomb grouper LHβ (AB525771.1) and FSHβ (AB52770.1) were used to design a series of gene-specific primers and probes in Primer Express (Thermo Fisher Scientific, Waltham, MA, USA). Their sequences are shown in Appendix A. The primers and probes were purchased from Integrated DNA Technologies, Inc. (Coralville, IA, USA). Assays were run on a Light cycler 480 (Roche Diagnostics, Mannheim, Germany) in 384-well plates (Roche Diagnostics, Mannheim, Germany), using standard cycling conditions: 95 °C for 10 min, followed by 45 cycles of 95 °C for 10 s and 58 °C for 30 s. The reaction volumes (20 µL) contained 5 µL of cDNA template, 0.5 µM of the forward and reverse primers, 0.2 µM of the probe, 10 µL of FastStart Essential DNA Probes Master 2 (Roche Diagnostics, Mannheim, Germany), and 4 µL of distilled water. We used a plasmid containing a partial *fshb* or *lhb* cDNA sequence as the standard for quantification (Appendix A). The standard sets of 10 points ranged from 1 × 10^10^ to 1 × 10^1^ copies and were prepared by 10 serial dilutions. Technical duplicates were run for all the experimental samples and standards.

### 2.8. Statistical Analysis

Owing to technical considerations, not all the data were statistically analyzed in the present study. The qPCR data are shown as the mean ± standard error, and statistical analyses were carried out with JMP 14.2 (SAS Institute Inc., Cary, NC, USA). Dunnett’s test was used to compare the treatments with the control at each sampling time to elucidate the effects of pheromonal cues and HCG on the levels of pituitary *fshb* and *lhb*.

## 3. Results

### 3.1. Spawning Grounds and Numbers of Aggregating Fish

The spawning aggregation of honeycomb groupers was concentrated along 150 m of the reef edge (Figure 1C). The number of individuals counted from 27 May to 4 June 2018 is shown in Table 1. In total, 16–21 territorial males were observed at the reef edge and its wall throughout the experimental period, except on 29 May, when observations could not be conducted. The number of females increased rapidly from 30 May, the day after the full moon, and 17–41 females were observed until 2 June. On 4 June, the number sharply decreased.

### 3.2. Grouper Behavior in the Observation Area

The results of observing the interactions of males with other males and females from 25 to 28 July 2018 and from 12 to 23 June and 13 to 15 July 2019 are shown in Figure 2. The dominant males stayed on the rocks of the wall and patrolled their territory. Aggressive bumping, chasing, and biting were occasionally observed between males. Females migrated to the aggregation site later than the males, after which they were generally observed within the male territories. Throughout this time, males would slowly approach and stay close to females. Then, they would quiver with a slightly open mouth for a few seconds. Individual males exhibited courtship behavior toward several females in their territory. Coloration patterns were observed depending on behavior: a honeycomb-dot pattern (Figure 2A,B) was normally observed, a striped pattern (Figure 2C,F) was observed when the fish were behaving aggressively or cautiously, and a white belly and eye bar (Figure 2D,F) were exhibited only by males during male-to-female courtship.

### 3.3. Ovarian Development in the Observation Area

The macroscopic changes in honeycomb grouper ovaries during aggregation are shown in Figure 4. The ovaries captured on 30 and 31 May were filled with pre-ovulatory oocytes (the white section of the ovaries, Figure 4A). Ovulatory eggs, which became transparent with hydration, appeared between 1 and 3 June (Figure 4B–D). The ovaries were considerably smaller on 4 June, although a small number of ovulatory eggs were observed in the shrunk ovaries. The GSI of pre-ovulatory females was <12 on 30 and 31 May (Figure 5). It increased to 15–25 in females with ovulatory eggs on 1 and 2 June and then decreased, reaching its minimum value on 4 June. The PPW decreased each day from 1 to 3 June.

### 3.4. Egg Collection in Captivity

The results of honeycomb grouper egg collection in the rearing experiment conducted in June 2016 are shown in Table 2. In the control females, which were reared separately from males, no eggs were collected during the experiment. However, in the F + M group, eggs were observed 72, 96, and 120 h after the start of the experiment. Eggs were also observed in the MW group 96 and 120 h after the start of the experiment. In July 2016, released eggs were collected from the MW tank 72 h after the start of the experiment, while no released eggs were observed in the control group (Appendix A).

The phenomenon of no ovulation in females without males was always observed throughout our experiment, even when different tanks were used in this experiment. This was also the case in the previous preliminary tests. Therefore, there is no influence of the tank or other rearing conditions.

### 3.5. Oocyte Stages after Induction of Ovulation in Captivity

The changes in oocyte development stage after control, F + M, and MW treatments are shown in Table 2. The females in the control group that were sampled prior to the start of the experiment showed completed vitellogenesis; however, no ovulatory oocytes were observed. After the experiment started, the oocytes remained in the full-grown stage in all control females and none transitioned to ovulatory oocytes. After 120 and 140 h, atretic oocytes were observed in several of the control females. In both the F + M and MW groups, ovulatory eggs were observed 72, 120, and 140 h after the start of the experiment. After 144 h, the ovulatory females had shrunken ovaries containing a small number of ovulatory eggs. In the replicate experiment, four females in the MW group completed ovulation. The other MW-treated females and the control females were suspended in the pre-ovulatory stage (Appendix A).

The changes in oocyte development stage after HCG and MW treatments are shown in Table 3. In the initial group, all the females were TY-I, and all the females in the control group remained TY-I throughout the experiment. In the HCG group, TY-II appeared at 24 h, and Mn and R appeared at 48 and 72 h, respectively. In the MW group, all females remained TY-I for 24 h. TY-II appeared after 48 h and Mn and R appeared after 72 h.

### 3.6. qPCR for Pituitary fshb and lhb Subunits

The qPCR results for *fshb* and *lhb* are shown in Figure 6. The expression level of *fshb* in the control and the HCG groups did not show significant changes throughout the experimental period. In the MW group, the *fshb* level tended to increase at 48 h; however, no significant differences were confirmed between the HCG and control groups (Dunnett’s test, *p* > 0.05), and the levels decreased again at 72 h.

The expression level of lhb in the control and the HCG groups did not show any trend during the experiment. The *lhb* levels of the MW females tended to increase at 48 and 72 h. However, there were no significant differences between the MW and control groups at any sampling time (Dunnett’s test, *p* > 0.05).

## 4. Discussion

We conducted this study to elucidate the ovulation and spawning characteristics of honeycomb groupers. This is the first report of the lunar cycle-associated aggregation of honeycomb groupers in spawning grounds on a natural coral reef. Moreover, we are the first to identify the potential involvement of male pheromonal cues in the maturation of female honeycomb grouper ovulation.

During previous observations of wild groupers along the coral reef edge in Nakijin, we temporarily observed many honeycomb groupers in a certain area, indicating that this site may be used as a spawning ground. Therefore, we conducted detailed surveys focused on this area. Consequently, honeycomb grouper aggregation was observed around the reef edge at a depth of 6 m during the spawning period. The spawning grounds of other coral reef groupers are often located in channels with strong water currents between islands or reefs [3,21,22]. The spawning ground identified in this study was located in a waterway extending from the mouth of a small river, with strong currents and tidal changes. Therefore, the honeycomb groupers probably selected this site because of its strong water current. However, the area of the aggregation site and the number of aggregating individuals tended to be smaller for the honeycomb grouper (up to 58 individuals) than for other grouper species (up to hundreds or thousands of individuals) [3,21,22,23,24]. It is thought that among grouper species, middle- and large-sized species tend to exhibit migration and aggregation [2]. The aggregation of small-sized groupers has previously only been observed in the white-streaked grouper (TL < 40 cm), with several kilometers (mean = 5.2 km) between their normal habitat and spawning grounds [3,25]. The honeycomb grouper (TL < 25 cm), which is smaller than the white-streaked grouper, is one of the smallest grouper species. The spawning ground of the honeycomb grouper was found at the edge of a coral reef, with an even shorter distance between their adjacent habitat and spawning ground than the white-streaked groupers displayed. It is possible that the choice of spawning ground depends on body size, and honeycomb groupers do not have the ability to migrate long distances. Therefore, the distance between the habitat and spawning grounds of honeycomb groupers is shorter than that of middle- and large-sized groupers.

We found that female and male honeycomb groupers exhibited different aggregation characteristics. The number of females increased rapidly, and aggregation was maintained from 1 to 5 days after the full moon. Interestingly, ovulatory eggs were observed in the aggregating females. Our results correspond to previous reports that female GSI peaks at full moon, and that they travel from their habitat areas in coral reefs or lagoons to the reef edge or beyond it [17,19]. Conversely, a large number of males (approximately 20) were observed at the aggregation site before the full moon, and this number was maintained until the females disappeared. This is similar to what has been observed in other aggregating groupers, such as the camouflage grouper (*E. polyphekadion*) [22] and white-streaked grouper [3]. The early migration of males to the spawning grounds may be linked to the territorial nature of groupers [22,26,27,28].

We also observed males exhibiting courtship behavior toward females at the spawning ground. After the females arrived at the spawning ground, the males visited them. After a few days of male–female interaction, the oocytes began to change, and ovulation was noted. In a study on aggregating *Plectropomus areolatus*, females entered male territories and male–female interactions occurred prior to spawning, with ovulation occurring a few hours prior to spawning [21]. Other groupers also exhibited male–female interactions in aggregation sites before spawning [4,28,29].

By rearing groupers in captivity, we confirmed that a male-released pheromone is involved in the final stage of oocyte maturation and ovulation in female honeycomb groupers. This explains why the males visited the females at the spawning grounds. In our artificial rearing experiment, ovulation was observed in females reared with males and in females reared in water from tanks containing males. Conversely, if females were not exposed to either males or male rearing water, vitellogenic oocytes remained in their ovaries. These results illustrate that female ovulation requires stimulation by a substance contained in the water, not by visual stimulation. In other words, male-derived cues were present in the male rearing water. These results strongly suggest that pheromones released by the males are involved in the spawning of female honeycomb groupers. The latency period of these ovulation-inducing pheromones was just three days in this rearing study. Thus, in the natural habitat, the females received pheromones from the males during male–female interactions in this phase, promoting physiological changes toward ovulation, and the three-day period is considered the time required for these changes to occur.

The effects of male pheromones in inducing female ovulation have been reported in several freshwater species [30,31,32,33]. However, no reports have been published on ovulation- and spawning-inducing pheromones in marine fish, although there have been studies indicating pheromone intervention in spawning. In *Sparus aurata* [34], the removal of males from spawning females had negative effects on spawning frequency, egg production in the ovary, and sex hormone levels associated with the reproductive axis. This study showed that the presence of males plays an important role in spawning, even in marine fish. Our study is the first to reveal that this male cue is probably a pheromone. Furthermore, our results show that FOM, ovulation, and spawning are not regulated only by the lunar cycle, despite previous findings that the reproductive cycle of female honeycomb groupers is synchronized with the lunar cycle [17,18,19]. It appears that lunar cycle-related cues regulate the oocyte growth phase in the early stage of oocyte development, whereas male pheromonal cues regulate the FOM and ovulation phases. It is possible that male pheromone production is also influenced by the lunar cycle.

Fish FOM is regulated by the HPG axis [12]. In particular, MIH synthesis and OMC acquisition by LH directly induce FOM during the final step of oogenesis [12]. However, the factors that induce this endocrine system and physiological action differ among species. Our current findings indicate that a male pheromone is a cue that stimulates the HPG axis for FOM in honeycomb groupers. Similarly, in rainbow trout (*Oncorhynchus mykiss*) [35] and sea lampreys (*Petromyzon marinus*) [36], pheromones play a role in inducing endocrine mechanisms related to the final maturation of the fish. In addition, steroid compounds in the urine of tiger groupers (*E. fuscoquttatus*) show pheromonal effects that induce sex steroids, although their relationship with maturation has not yet been clarified [37]. To date, there have been no reports that clearly show the relationship between pheromones and the endocrine system in FOM; however, the role of the HPG axis in FOM has been demonstrated in many teleost fish [12]. Generally, GtH, in particular LH, has two roles in stimulating the synthesis of MIH and the acquisition of OMC [12,13,14,15,16]. In this study, the GtH β-subunit expression levels in the pituitary glands of females treated with male rearing water were assessed. Consequently, some individual females with increased *lhβ* expression appeared after treatment with male rearing water for 48–72 h; however, the effect was not statistically significant. In order to make this result more reliable, we planned additional experiments in 2020. Unfortunately, the experiments were canceled because the COVID-19 pandemic started in Okinawa. However, an interesting result is that, in the MW group, the FOM process proceeded from 48 to 72 h after treatment. Similar changes were observed in the individuals treated with HCG. These results indicate that the OMC acquisition and FOM caused by GtH were induced by the male rearing water. Thus, we can assume that pheromones from males promote GtH synthesis, which is involved in FOM. FOM did not occur in the control group in which male rearing water was not used, strongly suggesting the presence of a pheromone in the male rearing water that induced OMC acquisition and FOM. Although these pheromone substances remain unidentified in this grouper, sulfated sex steroids and bile acid have been identified as sex-related pheromones in rainbow trout [35] and sea lampreys [36], respectively. We plan to focus on sex steroids and their metabolites, or other sex-related substances released by mature male groupers in our future work.

In this study, we clarified one aspect of the reproductive characteristics of the honeycomb grouper through field surveys and breeding experiments. However, the number of individuals caught in the field and used in the rearing experiments in this study is not sufficient. In order to provide a clearer scientific explanation, it is necessary to increase the number of individuals used in the study. However, on the other hand, to conduct research while protecting natural resources, it is necessary to keep the number of individuals to a minimum. In any case, we understand that there is a need for ongoing research to complement this study. Unfortunately, due to the impact of COVID-19, those studies have not been accomplished. This will be a future challenge.

## 5. Conclusions

In conclusion, our findings revealed that honeycomb groupers aggregated at a spawning ground at the edge of a coral reef lagoon. The males began to aggregate a few days before full moon and the females arrived 1–2 days after the full moon and stayed in the male territories. Male honeycomb groupers exhibited courtship behavior toward the females within their territory during the aggregation period, and females completed ovulation a few days after this male-to-female courtship. A male-to-female interaction via pheromones is necessary for the execution of spontaneous FOM and ovulation, stimulating GtH synthesis and OMC acquisition. Our results suggest that lunar cycle-related cues regulate early oocyte development, whereas male pheromonal cues synchronize the timing of FOM and spawning in this fish. Although it is difficult to obtain enough experimental fish while protecting natural resources, we will conduct experiments with an increased number of experimental fish and using new indicators, such as the expression of maturation-related genes. After the COVID-19 pandemic subsides, we plan to resume the experiments, which are currently suspended, in order to better understand the maturation mechanism.

## Figures and Tables

**Figure 1 cells-11-00484-f001:**
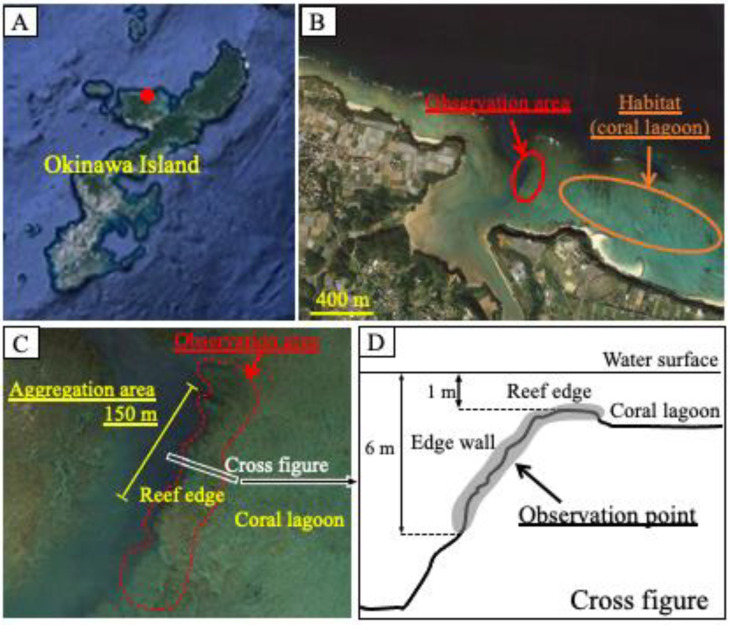
Maps of the observation site on Okinawa Island. (**A**,**B**) Locations of the observation site (red closed circle in (**A**)) and spawning ground (red open circle in (**B**)). (**C**) Aggregation area (approximately 150 m) in the spawning ground and observation area (red dotted line). (**D**) Cross-section of the spawning ground indicating the observation area (grey-colored area). These images were captured in Google Earth Pro 2018: http://www.google.com/earth/index.html (accessed on 24 November 2018).

**Figure 2 cells-11-00484-f002:**
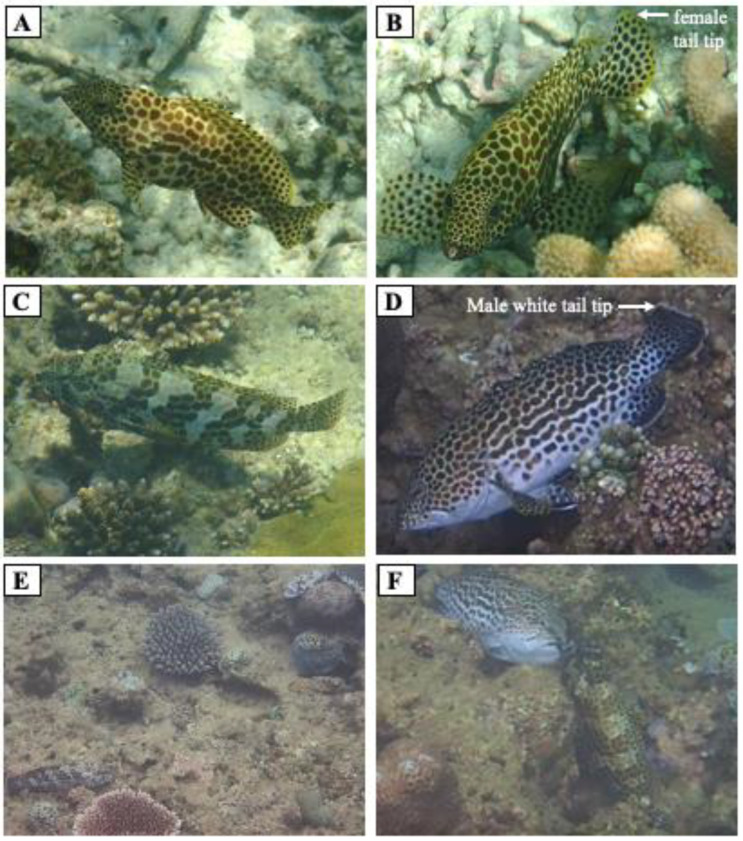
Body shape and color patterns exhibited by both sexes of honeycomb grouper during the daytime in the spawning ground. Females showed swollen abdomens with a normal honeycomb-dot color pattern (**A**,**B**). Males and females also exhibited a striped pattern (**C**). Males showed a white belly and white tail-tip (**D**). The striped pattern was observed in males when they exhibited aggressive behavior toward other males (**E**). The striped pattern was observed in females during male–female interactions, while males showed a white belly with an eye bar during male–female interactions (**F**).

**Figure 3 cells-11-00484-f003:**
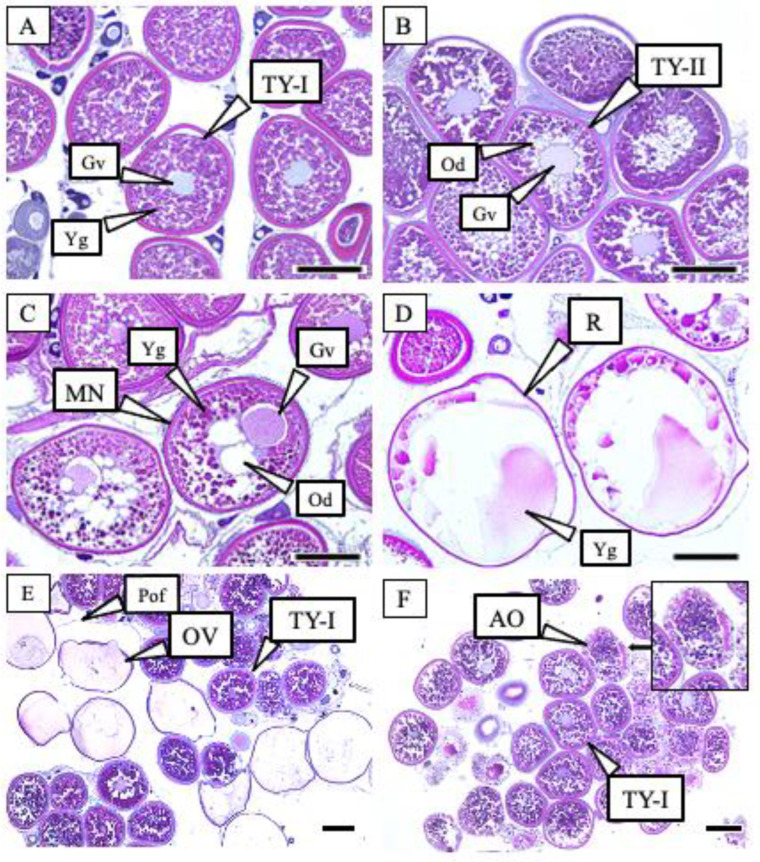
Oocyte development stage during final oocyte maturation. (**A**) Tertiary yolk stage-I (TY-I): the oocyte is filled with yolk globules (Yg) and the germinal vesicle (Gv) is located in the center of the cytoplasm. (**B**) Tertiary yolk stage-II (TY-II): the oocyte is filled with yolk globules and oil droplets (Od) are gathered around the germinal vesicle. (**C**) Migratory nucleus stage (MN): yolk globules and oil droplets are coalesced, and the germinal vesicles have migrated to the animal pole. (**D**) Ripe stage (R): coalesced yolk globules are hydrated, and germinal vesicles have broken down. (**E**) An ovulatory ovary of a female: ovulated eggs (OV) are released from the follicle and post-ovulatory follicles (Pof) and pre-ovulatory oocytes remain in the ovigerous lamella. (**F**) An ovary with atretic oocytes (AO): atretic oocytes appeared in females reared in artificial conditions at 120 and 144 h. Scale bar = 200 μm.

**Figure 4 cells-11-00484-f004:**
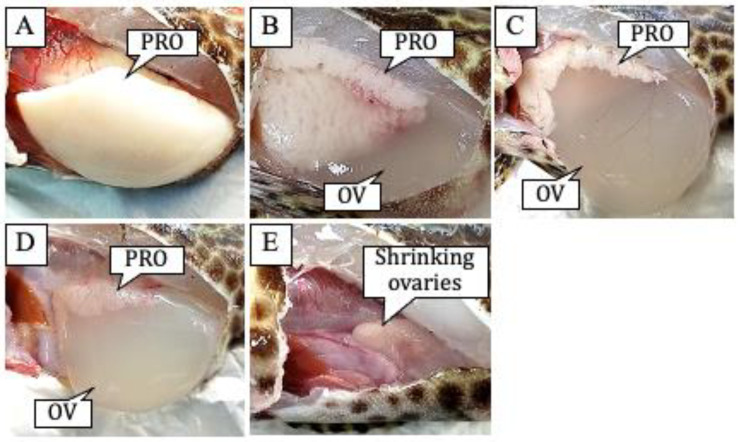
Morphological changes in the *Epinephelus merra* ovaries collected from the spawning aggregation site in May and June 2018. (**A**) An ovary of a pre-ovulatory female captured on 31 May. (**B**) An ovary of an ovulatory female captured on 1 June. (**C**) An ovary of an ovulatory female captured on 2 June. (**D**) An ovary of an ovulatory female captured on 3 June. (**E**) A shrinking ovary on 4 June. PRO indicates pre-ovulatory oocytes and OV indicates ovulated eggs.

**Figure 5 cells-11-00484-f005:**
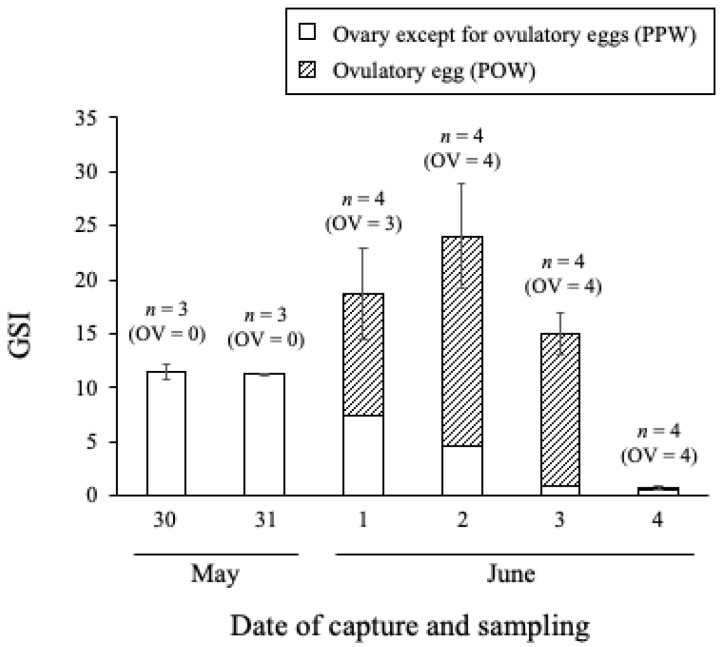
Gonadosomatic index of aggregating *Epinephelus merra* females in the observation area after the full moon. Open columns show the proportion of pre-ovulatory oocytes (PPW), and hatched columns show the proportion of ovulatory eggs (POW). Error bars indicate standard deviation, and the number of samples is shown as “*n*”. OV, number of ovulated individuals.

**Figure 6 cells-11-00484-f006:**
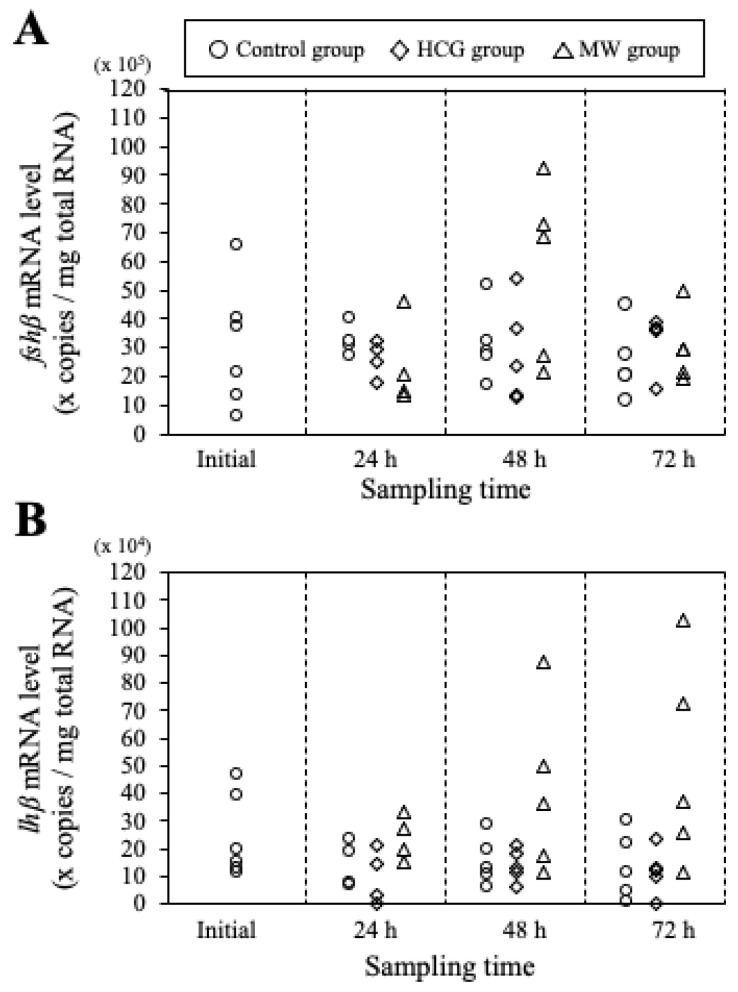
mRNA levels of GtH β-subunits, *fshβ* (**A**) and *lhβ* (**B**), in the pituitary. Y-axis shows the mRNA concentration normalized by total RNA. X-axis shows sampling time after the start of the experiment. The symbols indicate the experimental groups as follows: ○, control group; ◇, human chorionic gonadotropin (HCG)-treated group; △, male water (MW)-treated group.

**Table 1 cells-11-00484-t001:** Number of individuals in the spawning ground at Nakijin.

Date	May					June			
	27	28	29 *	30	31	1	2	3	4
M	19	18	-	18	16	19	21	21	18
F	2	3	-	17	29	39	41	19	7

* Full moon was on 29 May. The observation was not conducted on 29 May owing to heavy rain and unclear view caused by the river water.

**Table 2 cells-11-00484-t002:** Egg collection in rearing tanks and oocyte stages in females.

Groups [M:F]	*N*	Initial	24 h	48 h	72 h	96 h	120 h	144 h
Cont. [0:6]	24	-	-	-	-	-	-	-
		TY-I(6) *n* = 6	NI	NI	TY-I(6) *n* = 6	NI	TY-I(2), AO(4) *n* = 6	AO(6) *n* = 6
F + M [1:5]	15		-	-	+	+	+	-
			NI	NI	TY-I(2), OV(3) *n* = 5	NI	TY-I(3), OV(2) *n* = 5	TY-I(2), OV(3) *n* = 5
MW [0:6]	18		-	-	-	+	+	-
			NI	NI	TY-I(4), OV(2) *n* = 6	NI	TY-I(3), OV(3) *n* = 6	TY-I(4), OV(2) *n* = 6

-, no eggs in the rearing tank: +, eggs were collected. TY-I, tertiary yolk stage-I: OV, ovulatory egg stage: AO, atretic oocyte stage: *N*, the total number of females used in each experimental group: *n*, the number of sampled female at each sampling. The number in parentheses indicates the number of individuals in each oocyte stage. NI, no individual sampling.

**Table 3 cells-11-00484-t003:** The number of individual of in vivo ovarian development stage after treatment by FOM inducers.

Group	Time	*n*	Stage			
			Ty-I	Ty-II	Mn	R
Initial	0 h	6	6			
Cont.	24 h	6	6			
	48 h	6	6			
	72 h	6	6			
HCG	24 h	6	3	3		
	48 h	6	3		1	2
	72 h	6	1		1	3
MW	24 h	6	6			
	48 h	6	2	4		
	72 h	6	3		1	2

TY-I: tertiary yolk stage-I, TY-II: tertiary yolk stage-II, MN: migratory nucleus stage, and R: ripe stage. The number shows the number of individuals in each oocyte stage.

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
