# Peer review of "Male Pheromones Induce Ovulation in Female Honeycomb Groupers (Epinephelus merra): A Comprehensive Study of Spawning Aggregation Behavior and Ovarian Development"

_cells, 2022, doi:10.3390/cells11030484_

Round 1

Reviewer 1 Report

This manuscript should be considered as “preliminary” because all of its conclusions are based on inadequate sample sizes and very unclear statistical analyses.  For example, the field study simply serves to illustrate that this species engages in courtship and the females appear to ovulate at this time.  I am certain this applies to most fishes!  Certainly not an important result but does illustrate the authors understand some minimal information about the species under field conditions.  We are warned in lines 254-255 that the statistical analyses are minimal but no reason is given except, perhaps, the sample sizes are inadequate.  Inadequate sample sizes for whatever reason is simply unacceptable for any paper testing hypotheses. The minimal statistical analyses, such as those presented in Figure 5 have bars that have an N of either 3 or 4.  These small samples produced standard errors that are irrelevant in measuring variation.  A standard error based on such a small sample size cannot be so small!  I suspect they were using other numbers for their calculations.  Their most important experiment, about the possibility that females are stimulated by male pheromones, is presented without replication.  Both the number of eggs generated and the histological analyses are confusing because it is unclear how many females in this one replicate were affected by the “pheromones” and how many females produced eggs and the quantity of eggs produced by each female.  

Author Response

We think reviewer for careful reading our manuscript and for giving comments.

We revised the problems in every point based on your advice as follows.

Point 1

This manuscript should be considered as “preliminary” because all of its conclusions are based on inadequate sample sizes and very unclear statistical analyses. For example, the field study simply serves to illustrate that this species engages in courtship and the females appear to ovulate at this time. I am certain this applies to most fishes! Certainly not an important result but does illustrate the authors understand some minimal information about the species under field conditions.

 Response 1

As you pointed out, this study, which investigated spawning and associated physiological changes in the field, has a small sample size and weak scientific support. I am well aware of this, but it is extremely difficult to link spawning patterns in the field and physiological changes, and I think this study is valuable in understanding fish reproduction in this point.

You also pointed out that the communication between males and females on the spawning grounds and the associated phenomena of final maturation and ovulation may be observed in many fishes, but even this information is not sufficient for marine fishes. Also, even in aquarium experiments, spawning patterns and ovulation induction mechanisms vary greatly among fish species. The results observed in this study are important information when considering the reproduction of grouper. We believe that it is important to provide this information to our audience.

The statistics and repetition of experiments has been pointed out by other readers. We have added new explanations for some of the inadequacies.

Point 2

We are warned in lines 254- 255 that the statistical analyses are minimal but no reason is given except, perhaps, the sample sizes are inadequate. Inadequate sample sizes for whatever reason is simply unacceptable for any paper testing hypotheses. The minimal statistical analyses, such as those presented in Figure 5 have bars that have an N of either 3 or 4. These small samples produced standard errors that are irrelevant in measuring variation. A standard error based on such a small sample size cannot be so small! I suspect they were using other numbers for their calculations.

Response 2

As you pointed out, we understand that a small sample size is a problem in any experiment.

However, in conducting the experiment in the field while considering the sustainability of biological resources, we were forced to use the minimum number of fish.  The value of this study is to confirm the reproductive physiology of groupers in the field. Obtaining true reproductive information in a natural environment is necessary to truly understand the organism. I fully understand your point of view on this, and would appreciate your understanding.

Also, regarding the statistics, I used standard deviation instead of standard error. However, the reason for the small error and deviation, despite the small number of individuals used, is that there is little variation in the body size of individuals that mature and gather at the spawning grounds. We have not intentionally aligned the sizes of the mature individuals, nor have we used other data for statistical processing.

Point 3

Their most important experiment, about the possibility that females are stimulated by male pheromones, is presented without replication. Both the number of eggs generated and the histological analyses are confusing because it is unclear how many females in this one replicate were affected by the “pheromones” and how many females produced eggs and the quantity of eggs produced by each female.

Response 3

In this experiment, the exposure test with male rearing water was repeated three times.

There is a clear difference between the groups with and without male breeding water in the tanks.

There are seven females in each tank, and although we don't know which females spawned in the group with male rearing water, we are certain that spawning took place in this group.

We believe that this is a valid experiment to prove that pheromones exist in male breeding water.

Reviewer 2 Report

The paper is clear ; the subject of the research is innovative and interesting. The sampling protocol is well described. The stages of female oocyte maturation are described accurately and individually which brings a lot of information. 

I regret that the measurement of plasma levels of hormones (MIS, 11KT,..) are not done, because their involvement during ovulation is very informative and could make it possible to consider identifying the male pheromone involved in female stimulation.

The question of the synchronization of ovulations with the full moon, could be approached as a phenomenon related to the biological clock : the link between the regrouping of fish, the laying and the full moon, could it be perceptible (visible) for example by melatonin hormone levels, or genes from the clock transcripts?  This view could be considered for further analyses in addition to those presented here.

The PCR technique was applied to the pituitary gland. This could have been complemented by analyses of genes in the hypothalamus, kisspeptine or RFRP, known to control the release of GnRH, in relay of the melatonin hormone produced by the pineal gland.

Author Response

We think reviewer for your valuable comments. Thank you also for your understanding of our research.

Our comments on your advice are described as follows.

Point 1

The paper is clear ; the subject of the research is innovative and interesting. The sampling protocol is well described. The stages of female oocyte maturation are described accurately and individually which brings a lot of information.

I regret that the measurement of plasma levels of hormones (MIS, 11KT,..) are not done, because their involvement during ovulation is very informative and could make it possible to consider identifying the male pheromone involved in female stimulation.

The question of the synchronization of ovulations with the full moon, could be approached as a phenomenon related to the biological clock : the link between the regrouping of fish, the laying and the full moon, could it be perceptible (visible) for example by melatonin hormone levels, or genes from the clock transcripts? This view could be considered for further analyses in addition to those presented here.

The PCR technique was applied to the pituitary gland. This could have been complemented by analyses of genes in the hypothalamus, kisspeptine or RFRP, known to control the release of GnRH, in relay of the melatonin hormone produced by the pineal gland.

Response 1

There are limitations to physiological experiments using grouper fishes in field studies. However, we believe that the ecophysiological perspective is the most important to know the reproduction in fish

As you mentioned, measuring sex steroids is a important point. In this case, due to sampling circumstances, we were unable to measure the steroid in the blood. We are currently preparing to study the changes in sex steroids using rearing individuals.

We have already completed preparations to measure gene expression of GnRH, Kiss and other genes in grouper fish and are also planning to establish a measurement system for various factors, such as melatonin and clock genes.

Reviewer 3 Report

Honeycomb groupers, Epinephelus merra, were studied in the field and in captivity. In the field study the aggregation of groupers at the spawning group were studied over time as was the ovarian development at different phases. In the tanks the effects of males in the tanks and water from tanks containing males on female ovarian development, ovulation and other reproductive parameters were studied. Both the presence of males and male water stimulated ovarian group and ovulation, i.e. there appeared to be an effect of male pheromones on the females, something which is new for marine fishes, though well known for freshwater fish. HCG injections also stimulated maturation in females, which appeared clear but is of little scientific news value.

A strong limitation of the study is the almost total absence of statistics. It is really only used for the mRNA levels of fsh-beta and lh-beta in the pituitaries in females treated with male water, HCG or controls (Figure 6), which were not significant.

Even in the absence of statistics it appears that male water stimulates ovarian maturation and there are three experiments showing this (if I understand it right, it is not entirely clear) but a possible tank effect may be present and this possibility should at least be commented. All females in each treatment were in the same tank and if one female started to mature, she could have influenced the others so that they also matured, thus each individual is not a completely independent unit.  

Other comments:

Line 113. “The fish were easily identified”  I guess that this means that they were easily identified to species and sex, but it could also mean that they were individuals were identified. Write this more clearly.

Line 218. Why was the HCG solution mixed with cocoa butter?

  1. 6 HCG injections. Were there any control injections?

  1. 6. Any environmental enrichment in the tanks? How were the fish fed?

  1. 6. How many experiments with male water were conducted? One that started on 9 June and the results of which are shown in table 2, another parallel to the HCG treatment shown in Table 3 and a third shown in one of the supplement tables? The last one (if it is a separate experiment, the numbers do not fit the others anyway) appears to be absent in the M & M.

Figure 4. “Morphological changes in the oocytes” The title is not well chosen, it shows changes in the ovary, not in the oocytes that are too small to be seen.

Figure 5. It is not clear to me what the figure shows. Are all fish at a certain date in a similar state, i.e. are all ovulatory or none? This must be clarified, and if that is not the case, the data need to presented in another way. In the captivity experiment (Table 2) all fish are not at the same stage at all times. What is the meaning of the “SE < 0.01” and “SE < 0.2” on two of the bars?

Table 2 (and M & M) above, line 227 “every 24 hours sampling in each experimental group (5 females) was continued still 72h (dubious English and not clear to me how many females were used in each treatment. If I understand the table correctly there 6 females sampled at 72h, 6 at 120h and 6 at 144h in the MW group or 18 altogether.

Author Response

We think reviewer for careful reading our manuscript and for giving comments.

We revised the problems in every point based on your advice as follows.

Point 1

Honeycomb groupers, Epinephelus merra, were studied in the field and in captivity. In the field study the aggregation of groupers at the spawning group were studied over time as was the ovarian development at different phases. In the tanks the effects of males in the tanks and water from tanks containing males on female ovarian development, ovulation and other reproductive parameters were studied. Both the presence of males and male water stimulated ovarian group and ovulation, i.e. there appeared to be an effect of male pheromones on the females, something which is new for marine fishes, though well known for freshwater fish. HCG injections also stimulated maturation in females, which appeared clear but is of little scientific news value.

A strong limitation of the study is the almost total absence of statistics. It is really only used for the mRNA levels of fsh-beta and lh-beta in the pituitaries in females treated with male water, HCG or controls (Figure 6), which were not significant.

Response 1

As you pointed out, I accept that there is little scientific support by statistics. However, even in the field of physiology, histological changes and clear gonadal development and spawning phenomena can be discussed as facts without using statistics. In recent years, there has been a push to evaluate such phenomena without depending on statistics.

In addition, this study uses a minimum number of individuals in consideration of the sustainability of natural resources. That is also one of the reasons for your point.

We explain about the results of GtH gene expression. There is certainly no significant difference in each group. This is because there is a lot of individual variability. Unlike experimental model fish such as killifish and zebrafish, these results often happen using the fish corrected from naturally. However, h-beta tended to increase by male water treatment. Additionally, ovulation was induced by administration of gth in HCG group. These results suggests GtHs are involved in male-water-inducing ovulation.

Our goal is to understand the phenomena that occur in the natural field as physiologically as possible. We are well aware of your points, but we strongly feel that it is important to widely share the contents of this study.

Point 2

Even in the absence of statistics it appears that male water stimulates ovarian maturation and there are three experiments showing this (if I understand it right, it is not entirely clear) but a possible tank effect may be present and this possibility should at least be commented. All females in each treatment were in the same tank and if one female started to mature, she could have influenced the others so that they also matured, thus each individual is not a completely independent unit.

Response 2

We consider the possibility of tank influence to be non-existent. We also believe that there is no influence of females that have started to spawn. The reason for this idea is that ovulation was induced only in the male rearing water group, in both our experiment to determine whether or not spawning occurs and the experiment to HCG treatment and determine GtH gene expression.

A replicated experiment to confirm the effect of male water was conducted and is described in L172-178 (Methods) and L320-322 (Results).

In addition, we have conducted several similar tests in the past, and the same results have been obtained in those experiments. It is reproducible even with different tanks and in different breeding facilities. However, these have not yet been published in a paper.

We have added the following text; “The phenomenon of no ovulation in female without males was always observed throughout our experiment, even when different tanks were used in this experiment. This was also the case in the previous preliminary tests. Therefore, there is no influence of the tank or other rearing conditions.” (L323-326)

Point 3

Line 113. “The fish were easily identified” I guess that this means that they were easily identified to species and sex, but it could also mean that they were individuals were identified. Write this more clearly.

Response 3

This means that "the difference between males and females was easily distinguishable," not that it explains individual identification. The following sentence explains that "the females had a swollen abdomen and a yellow fin tip, while the males had a white fin tip".

Point 4

Line 218. Why was the HCG solution mixed with cocoa butter?

Response 4

We used cocoa butter as the adjuvant for maintain the plasma HCG level in the experimental fish during the experiment. In this experiment, we need to maintain the effect of HCG for several days in order to compare effects of HCG treatment with that of male water treatment. This method is one of the most common methods of administration of HCG or other hormones.

Point 5

1.6. HCG injections. Were there any control injections?

Response 5

The same dose of cocoa butter was injected in control group. This has been added to the manuscript (L228).

Point 6

  1. 6. Any environmental enrichment in the tanks? How were the fish fed?

Response 6

Each tank of all experimental groups contained several shelters to reduce stress of experimental fish.

Before start of the experiment, experimental fish were fed the silver-stripe round herring Spratelloides gracilis in stock tanks. During the experiment, we did not feed any baits.

This information has been added in the manuscript(L154, L179-181, L216-219).

Point 7

  1. 6. How many experiments with male water were conducted? One that started on 9 June and the results of which are shown in table 2, another parallel to the HCG treatment shown in Table 3 and a third shown in one of the supplement tables? The last one (if it is a separate experiment, the numbers do not fit the others anyway) appears to be absent in the M & M.

Response 7

We conducted male water exposure test 3 times. First time; materials and methods are written in line 150 – 171, results are written in line 316 – and table 2. Second time; M&M are written in line 172 - 178, results are written in line 320 – and supplementary data 2. Third time; M&M are written in line 213 - 237, results are written in line 342 – and Table 3.

The second experiment is shown as supplementary data, but was incorrectly described as data not shown in the text. It has been corrected (L322).

Point 8

Figure 4. “Morphological changes in the oocytes” The title is not well chosen, it shows changes in the ovary, not in the oocytes that are too small to be seen.

Response 8

As you suggested, we removed “oocyte” and changed the description (L304-305, L631-632).

Point 9

Figure 5. It is not clear to me what the figure shows. Are all fish at a certain date in a similar state, i.e. are all ovulatory or none? This must be clarified, and if that is not the case, the data need to presented in another way. In the captivity experiment (Table 2) all fish are not at the same stage at all times. What is the meaning of the “SE < 0.01” and “SE < 0.2” on two of the bars?

Response 9

In individuals who did not ovulate, GSI was calculated based on the weight of the ovaries. In ovulated fish, the weight of the ovulated area and the weight of the other part of the ovary were measured respectively, and the GSI for each weight was calculated, and then the average value was calculated.

Methods for calculation are described in L144. We have added here a method of expression for GSI (L145-148).

The number of ovulated individuals has been added to the figure.

Also, this figure has been changed from standard error to standard deviation.

SE indicates the standard error. In the figure, due to the small value of SE, it was shown as a number. However, it has been removed because it is misleading.

Point 10

Table 2 (and M & M) above, line 227 “every 24 hours sampling in each experimental group (5 females) was continued still 72h (dubious English and not clear to me how many females were used in each treatment. If I understand the table correctly there 6 females sampled at 72h, 6 at 120h and 6 at 144h in the MW group or 18 altogether.

Response 10

I'm sorry for the confusing expression.

In the control group, we first sampled 6 females as initial sampling at the beginning of the experiment. After that, we sampled 6 females at each time (at 72 hours, 120 hours, and 144 hours). The total is 24 fish.

In the MW group, we sampled 6 females at each time (at 72, 120, and 144 hours). The total is 18 fish.

However, in the group where males and females were kept at a ratio of 1:5 (M+F group), 5 females were sampled for each hour. The total was 15 fish.

We have revised the text description (L166-169) and Table 2 and its description (L650-651).